# communications
# engineering

## PERSPECTIVE

# Space bioprocess engineering on the horizon

Aaron J. Berliner [1,2✉], Isaac Lipsky[1,2], Davian Ho[1,2], Jacob M. Hilzinger[1,2],
Gretchen Vengerova [1,2], Georgios Makrygiorgos[1,3], Matthew J. McNulty[1,4],
Kevin Yates[1,4], Nils J. H. Averesch [1,5], Charles S. Cockell[1,6], Tyler Wallentine[1,7],
Lance C. Seefeldt[1,7], Craig S. Criddle[1,5], Somen Nandi [1,4,8],
Karen A. McDonald[1,4], Amor A. Menezes [1,9], Ali Mesbah[1,3] &
Adam P. Arkin [1,2✉]

Space bioprocess engineering (SBE) is an emerging multi-disciplinary field to design, realize, and manage biologically-driven technologies specifically with the goal of supporting life on long term space missions. SBE considers synthetic biology and bioprocess engineering under the extreme constraints of the conditions of space. A coherent strategy for the long term development of this field is lacking. In this Perspective, we describe the need for an expanded mandate to explore biotechnological needs of the future missions. We then identify several key parameters—metrics, deployment, and training—which together form a pathway towards the successful development and implementation of SBE technologies of the future.

Biotechnologies may have mass, power, and volume advantages compared to abiotic approaches for critical mission elements for long-term crewed space exploration[1,2]. While there has been progress in the demonstration and evaluation of these benefits for specific examples in this field such as for food production, and waste recycling, there is only just emerging possible consensus on the scope of the application of biosynthetic and bio-transformative technologies to space exploration. Additionally, there is almost no formal definition of the scope, performance needs and metrics, and technology development cycle for these systems. It is time to formally establish the field of space bioprocess engineering (SBE) to build this nascent community, train the workforce and develop the critical technologies for planned deep-space missions. SBE (Fig. 1a) borrows elements from a number of related fields such as the synthetic biology design process from Bioengineering, astronaut sustainability[3,4] and mission design from Astronautics[5,6], environmental-context, and constraints from the Space Sciences, and living systems habitability and distribution concepts from Astrobiology[7]. SBE represents an extension of the standard astronautics paradigm in meeting NASA's Space Technology Grand Challenges (STGCs) for expanding the human presence in space, managing resources in space, and enabling transformative space exploration and scientific discovery[8,9] (Fig. 1b). Aspirational realizations of SBE would feature prominently in establishment of in-orbit test-facilities, interplanetary waystations, lunar habitats, and a biomanufactory on the surface of Mars[10]. Differentiated from traditional efforts in space systems engineering, these SBE systems would encapsulate elements from in situ resource utilization (ISRU) for the production of biological feedstocks such as fixed carbon and nitrogen for use as inputs for plant, fungal, and microbial production systems[11,12], fertilizers for downstream use by plants[13]; in situ (bio)manufacturing (ISM) to produce materials

[1]Center for the Utilization of Biological Engineering in Space (CUBES), Berkeley, CA, USA. [2]Department of Bioengineering, University of California Berkeley, Berkeley, CA, USA. [3]Department of Chemical and Biomolecular Engineering, University of California Berkeley, Berkeley, CA, USA. [4]Department of Chemical Engineering, University of California, Davis, Davis, CA, USA. [5]Department of Civil and Environmental Engineering, Stanford University, Stanford, CA, USA. [6]UK Centre for Astrobiology, School of Physics and Astronomy, University of Edinburgh, Edinburgh, UK. [7]Department of Chemistry and Biochemistry, Utah State University, Logan, UT, USA. [8]Global HealthShare Initiative, Davis, CA, USA. [9]Department of Mechanical and Aerospace Engineering, University of Florida, Gainesville, FL, USA. ✉email: aaron.berliner@berkeley.edu; aparkin@lbl.gov

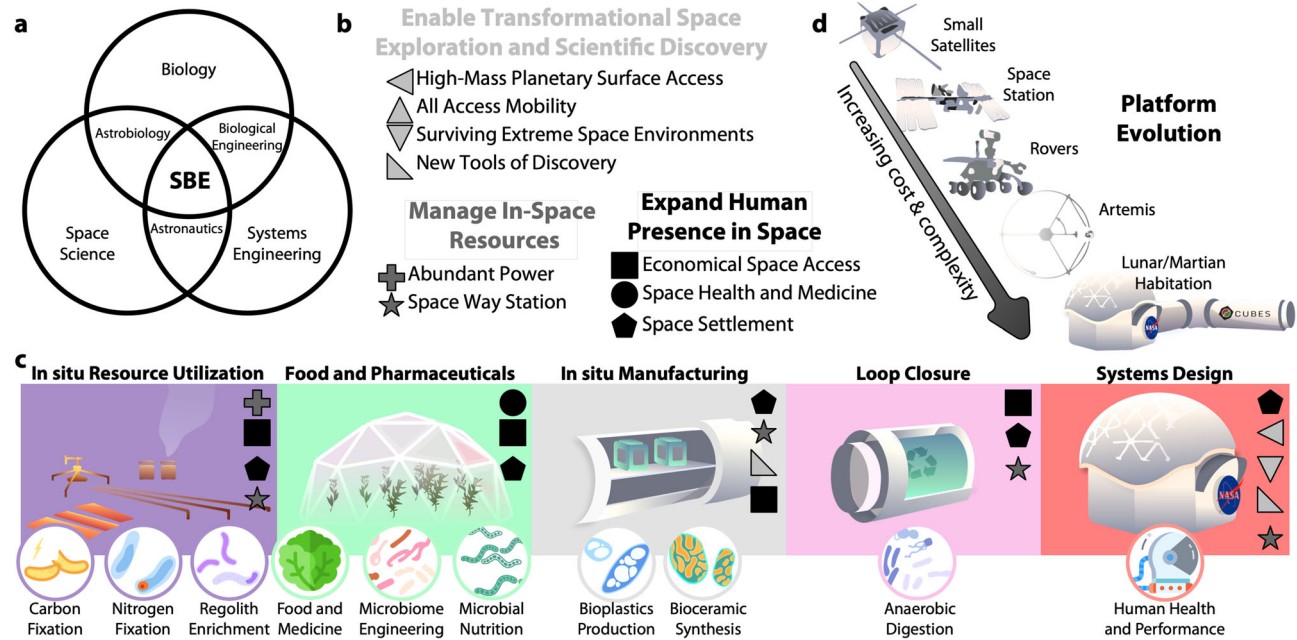

**Fig. 1 Overview of space bioprocess engineering challenges, components, and platforms. a** Venn Diagram-based definition of Space Bioprocess Engineering (SBE) as an interdisciplinary field. **b** NASA's space technology grand challenges[8] key by shape and colored by group. **c** Possible SBE components separated by colors for in situ resource utilization (ISRU), food and pharmaceutical synthesis (FPS), in situ manufacturing (ISM), and loop closure (LC), with the biological processes inherent to each represented below in circles. **d** Platform evolution for biological experiments starting with Earth-orbit CubeSats and proceeding through the ISS, Mars-and-Luna-based rovers, to Lunar and cis-Lunar based human and autonomous systems via the Artemis program.

requisite to forge useful tools and replacement parts[14], food and pharmaceutical synthesis (FPS) via plant, fungal and microbial engineering for increased productivity and resilience in space conditions, production of nutrients and protective/therapeutic agents for sustaining healthy astronauts[15,16]; and life-support loop closure (LC) for minimizing waste and regenerating life-support functions and biomanufacturing. Maximizing the productivity of the biomanufacturing elements increases the delivery-independent operating time of a biofoundry in space while minimizing cost and risk[17]. (Fig. 1c). Ultimately, efforts must be mounted to: (1) update the mandate to include SBE as a tool for enabling human exploration; (2) specialize the metrics and methods that guide SBE technology life-cycle and development; (3) further develop the means by which SBE technologies are designed for ground-based testing and matured on offworld platforms (Fig. 1d); and (4) train the minds entering the spacefaring workforce to better understand the leverage the SBE advantages and capabilities.

## An inclusive mandate to leverage SBE

While previous strategic surveys such as NASA's Journey to Mars program[18] and the 2018 Biological and Physical Sciences (BPS) Decadal Survey[19] have acknowledged that plants and microbes may be integral parts of life support and recycling systems, but can present challenges to the environmental operation of engineering systems in space due to contamination and other inherent drawbacks. However, no such survey has coherently called for the development of science and technology to engineer these organisms and their biotransformative processes in support of space exploration. The SBE community requires a mandate that identifies mission designs and elements for which engineering biosystems would be most appropriate, and defines the productivity, risk, and efficiency targets for these systems in an integrated context with other mission elements and in fair comparison to abiotic approaches. This will require integration of SBE

resources and knowledge across government, industry, and academia. Previous biological strategies should now specifically call for (1) definition of the physical engineering constraints on the production systems and development of optimized reactor/processing systems for these elements; (2) quantitative assessment of the bioengineering required to meet performance goals in space given the special physiology required in an offworld environment; and (3) development of efficient tooling for offworld genetic engineering along with the proper containment and clean-up protocols.

Such a mandate would result in: (1) a deeper, more mechanistic understanding of the growth and phenotypic characteristics of organisms operating in space-based bioprocesses taking into account issues of differences in gravity, radiation, light, water quality; (2) new applications of these organisms off-planet; (3) new reactors, bioprocess control designs and product processing/delivery technologies accounting for these conditions and the specific constraints of scaling and operational simplicity in space. The development of open, publicly accessible data and tools would enable rigorous comparison among biotechnologies and abiotic (physical and chemical) approaches, and across mission-scenarios of higher-fidelity. Ideally, this should create interative sub-communities that may collaborate and compete on different approaches to meet bioengineering goals and metricize results against the mission specifications.

SBE is an emerging engineering discipline and there are long but feasible routes from discovery, through invention to application. Furthermore, SBE is multidisciplinary and its utility within the larger space community demands specialized cross-training of diverse teams. In such situations, agencies like the Department of Energy have found it effective to ensure there is specific funding to support longer-term team science to accomplish ambitious scientific and technical goals. The Industrial Assessment Centers (IACs) program is one of the longest-running Department of Energy programs (started in 1976) and

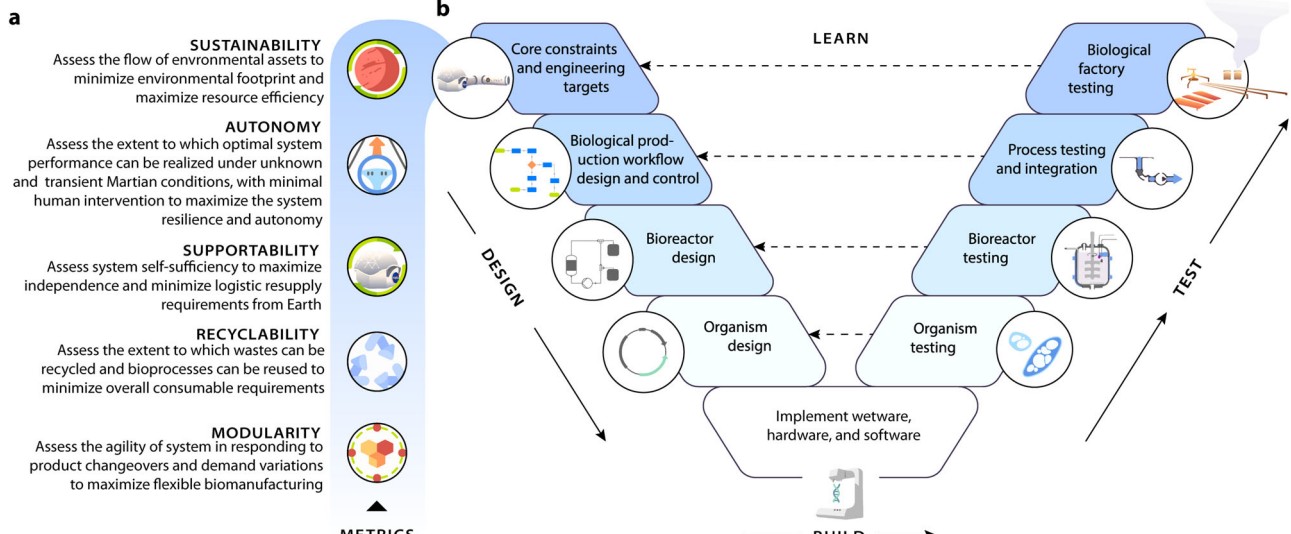

**Fig. 2 Overview of space systems bioengineering (SBE) performance metrics and the SBE-specific Design, Build, Test, Learn (DBTL) cycle.** The SBE performance metrics in **a** are shaded to correspond to the top level core constraints and engineering targets within the **b** DBTL cycle.

has provided nearly 20,000 no-cost assessments for small- and medium-sized manufacturers and more than 147,000 recommendations in an effort to reduce greenhouse gas emissions without compromising U.S. manufacturing's competitive edge globally[20]. Conversely, successful examples for demonstrating the effect of fostering multidisciplinary centers for space-based biotechnology can be found in NASA's Center for the Utilization of Biological Engineering in Space (CUBES), or ESA's Micro-Ecological Life Support System Alternative (MELiSSA) program —with the capabilities to design, prototype, and ultimately translate biological technologies to space while training the necessary workforce. Such centers are tasked with the development of initial concept trade studies; defining requirements; managing life-support interfaces; evaluating ground integration, operations, and maintenance; coordinating mission operations; and supporting and sustaining engineering and logistics[21,22]. However, these programs are generally restricted to shorter operation timelines—and would benefit from a longer horizon. This is especially true for SBE as biological developments generally require a longer timeframe for integration in industrial endeavors.

## Specialization of SBE metrics and methods

Response to the proposed expanded mandate above requires careful consideration of the space-specific performance metrics that SBE must fulfill. Payload volume, mass, and power requirements are made as small as possible and are limited in envelope by their carrier system. One of the most compelling aspects of biotechnology is the ability of such systems to adapt to these constraints relative to certain industrial alternatives. To efficiently evaluate and deploy novel biotechnologies, SBE experiments should begin with standardized unit operations that clearly define the desired biological function. This allows for a standardized experimental framework to test modular biotechnologies not only within the system to be engineered, but also within and between research groups. To define the minimal basis set of unit operations for a given mission, test and optimize the biotechnologies for each unit operation, and integrate each unit operation into a stable system, we propose to adopt the methods from standard bioengineering in the form of a Design-Build-Test-Learn (DBTL) cycle[23] (Fig. 2).

**Performance metrics**. The design phase of the DBTL cycle begins with the establishment of core constraints and engineering targets that can be explored by standardizing the high-priority performance metrics ({Modularity, Recyclability, Supportability, Autonomy, Sustainability})—which we argue gain special weight in space—from which downstream technoeconomic and life-cycle analysis decisions can be explored (Fig. 2a). The space-specific constraints on performance include: (1) an exceptionally strong weighting on a low mass/volume/power footprint for the integrated bioprocess; (2) limited logistic supply of materials and a narrow band of specifically chosen feedstocks; (3) added emphasis on simplicity of set-up, operation and autonomous function to free up astronaut time; (4) mission-context de-risking against cascading failure; (5) strong requirements for efficiency and closed-loop function to maximize efficient resource use and minimize waste products; (5) a critical need for modularity and 'maintainability' so that parts can be swapped easily, new functions added easily, and repairs can be done without logistical support beyond the crew; (6) an increased dependence on other mission elements such as provision of water, gases, astronaut wastes, power, and other raw materials such a regolith which may vary in abundance, quality, and composition in unpredictable ways; (7) the need to design sustainable and supportable operation across long time horizons without logistical support beyond the bounds of the local mission; (8) increased ability to operate in more extreme environments including low gravity, high radiation, low nutrient input, and other stressors; (9) process compatibility among common media and operational modes to allow for easy process integration and risk-reduction through redundancy of systems; and (10) further consideration and development of biocontainment of engineered organisms to prevent (or at least mitigate) unexpected dispersal of unwanted living systems in pristine or tightly controlled environments[24–26].

Ideally, this combination of performance metrics provides informative constraints on biology and technology choices. Feedstock, loop-closure, environmental parameters and product needs will constrain the minimal set of organisms to develop and test for growth rate, optimal cultivation, robustness and resilience to space conditions and shelf-life, safety and genetic tractability, product yield, titer and rate, feedstock utilization, ease of biocontainment, streamlining of purification, and waste streams[27]. Once suitable chassis organisms have been evaluated

and selected, the DBTL cycle can integrate staged co-design of the optimal process hardware (e.g., molecular biological set-ups, genetic engineering tools, bioreactors, and product post-processing systems) configuration, operating parameters, and process controllers. Aerobic organisms may be much more efficient but only viable in systems in which oxygen is available and easily obtainable. This in particular provides insight into the specific questions that require further study in terms of organism engineering. The question of anaerobic versus aerobic metabolism really depends on the product and the style of process—at small scale aerobics may have an advantage in terms of yield and rate, due to more energy being derived from the transfer of reducing equivalents to cellular metabolism—while at large scale, mass-transfer limitations are dominating these parameters (yield and rate), which gives anaerobics an advantage[28]. Additionally, bioproduct isolation and purification processes need to be considered beyond the Earth-centric means of fermentation. For example, cell-free bioproduction systems may prove critical in biotransformation and point-of-care biosensing as shown in recent space pharming techoeconomic analyses[29]. Operation of the cycle over increasing scale and ever more realistic deployment environments permits controlled traversal of the technology readiness levels for each technology and mission.

**Design-build-test-learn**. In the design phase, we argue that efforts must be made to (1) create a database of engineering targets (products, production rates, production yields, production titers, risk factors, waste/recyclability factors, material costs, operational costs, weight, power demand/generation) that set the core constraints for workflow and mission optimization; (2) leverage emerging pathway design software and knowledge bases[30] to identify the key types of biological production workflows (i.e., metabolic engineering strategies[31]) that need to be modified for different space-based scenarios; (3) identify the supporting bio-manufactory design elements within which these production workflows could be implemented[32–34]; and (4) identify the chassis organisms and other biological components[35–37] that will be required to compose the complete set for downstream engineering specifications. Systems designed from a minimal set of reliable parts, standard interconnects, and common controller languages also offer the best possible chance of characterized reliability under changing environmental conditions. Therefore, control of hardware and wetware should be augmented through the design and operation of software support. We see a fundamental effort in SBE as the amalgamation of space-driven hardware, software, and wetware that follows a synthetic biology DBTL cycle[38].

The foundation of new SBE performance metrics that guide the design phase of the DBTL cycle must be augmented with additional downstream efforts in the build and test phases to (1) develop a process design framework that takes in specific production needs in amounts/time over acceptable ranges under the constraints expected across different offworld scenarios; (2) create the biological, process, and mission design software platforms to allow sophisticated DBTL, risk assessment, and mission choice support; (3) create the sensor/controller sets that will allow real-time optimization of biological production workflows; and (4) develop the online process controller framework that coordinates reactor conditions and inter-reactor flows to optimize reliable production across all units within acceptable ranges with minimal power and risk. The realization of this SBE DBTL cycle depends on the integration of such benchmark models and modeling standards. These benchmarks describe the dynamics of all SBE processes and relate to the SBE metrics in the design phase from which optimization can be carried out in the learn phase.

DBTL cycles within the scope of SBE must prepare for both ground- and flight-based system operations. Ground-based developments must prioritize designs that meet the requirements for flight-based testing, during which system behaviors may be better characterized in unique environments such as those offered in micro- and zero-gravity. For instance, a biological nitrogen-fixing system on Earth must at least be designed to meet the mass and volumetric constraints required for validated ground-based simulators of microgravity, galactic cosmic radiation, and other physical stressors. Meeting certain requirements for time, power, and substrate usage is essential for any degree of long-term operation. This allows for the in-flight testing of bioreactors previously evaluated on Earth that can more directly measure the effects micro-gravity, radiation, and other stressors on the bioprocessing system. A combination of ground- and flight-based tests are required for the development of functional and robust space biosystems.

### Development of means for SBE flight
Deployment of SBE platforms as mission critical elements will likely be reserved for longer duration human exploration missions such as those in the Artemis or Mars programs[10]. These future programs are still in the concept and planning stage in development, but will certainly be composed of a myriad of technologies that range in degree of flight-readiness as standardized by NASA's Technology Readiness Level[39] (TRL, used to rate the maturity of a given technology during the acquisition phase of a program). Recent updates in NASA's definitions of and best-practices for applying the TRL paradigm led to the standardization and merging of exit criteria between hardware and software systems[40]. However, the TRL concept as it relates to SBE must be further expanded to include definitions and exit criteria for 'wetware' in addition and in relationship to hardware and software elements.

Deployment of SBE in space requires a level of rigor in technology acceptance that is of a different order than most Earth-based systems because mission failures are exceptionally costly and difficult to recover from. The missions into which SBE processes will integrate are hugely complicated and as noted above will be interdependent in complex ways. Thus while low levels TRLs can be reached through unit testing in modest formats both on Earth and in limited flight experiments, the integrated nature of the bioprocess control and engineering will require integration testing even at the TRL 4 and 5 levels[40]. To meet acceptance at TRL 6 and beyond will require long term planning realistic integration and deployment testing with actual sophisticated space missions and their logistics.

Even at low TRLs, research on the timescales needed to validate extended-use systems as would be leveraged on extended-stay forward deployment such as Martian or lunar missions are not possible given the current ISS capabilities and constraints. Constraints in astronaut time and limitations in hardware designed for shorter experiments prevent testing times comparable to long duration missions. Table 1 outlines a number of constraints on past and current experimental platforms and provides some basis for constraints of future systems (Fig. 1d). Here we note that extended multigenerational studies, especially in microbiology, can be difficult with some of the operational lifetimes[41]. Volume is also constrained, and available space is broken up into segmented rack testbeds and independent machines, which can prevent aspects of a system from interacting with each other (Table 1). Much of the testing hardware on the ISS is designed for front-end processing and basic science. Experiments in microbial observation[42,43], hybrid life support[44], and antibiotic response[45] require returning samples to Earth for efficient processing,

**Table 1 Constraints on past and current experimental platforms including small satellites, space stations, rovers, planned lunar habitation, and martian habitation.**

| Platform | Volume | Power | Op. lifetime | Temperature | Air comp. |
|---|---|---|---|---|---|
| CubeSat | 0.0187 m³ | 20–45 W | ~20 years | Requires heating unit within constraints | Self-contained |
| PocketQube | 0.000125 m³ | Variable | ~5 years | | |
| Bioculture System | Not stated | 140 W | ~60 days | 37–45 °C in main chamber, ambient to 5C in cooling chamber | Self-contained medical grade gas |
| WetLab-2 (SmartCycler) | 235.97 m³ | 350 W | Extractions <3 h, no lifetime stated | 50–95 °C | None, reliant on cabin air system |
| Rodent Habitat Hardware System | 0.019 m³ | Not stated | ~30 day experiments | Ambient temp, no heating module | |
| Compact Science Experiment Module | 0.0015 m³ | 3.2 W | >1 month experiments | | |
| Vegetable Production System (Veggie) | 0.48 m³ growth area | Not stated | >12 day experiments, can replace crops | | |
| Advanced Plant Habitat (APH) | 889.44 m³ growth area | | ~1 year | 18–30 °C | Self-contained gas supply |
| Spectrum | 10 × 12.7 cm internal area | | 12 day experiments | 18–37 °C | None, reliant on cabin air comp |
| BRIC-60 | 11.03 m³ | Unpowered | >12 day experiments | Ambient temp, no heating module | 60 M variant can draw from an external gas tank |
| BRIC-100 | 38.78 m³ | | | | Self-contained gas canister of designated composition |
| BRIC-100VC | 16.33 m³ | | 4.5 months | | |
| KSC Fixation Tubes (KFTs) | 0.2387 m³ | | 67 days | | |
| miniPCR | 0.00066 m³ | 65 W | ~2 year | <120 °C | Airtight, reliant on cabin air comp |
| Group Activation Pack-Fluid Processing Apparatus (GAP-FPA) | Eight 6.5 cm³ test tubes | Unpowered for manual | Not stated | 4–37 °C | |
| Multi-use Variable-g Platform (MVP) | Twelve 800 cm³ modules | Not stated | | 14–40 °C | |
| MinION | 0.0796 m³ | 5 W | ~1 year | Ambient temp, no heating module | |
| Perseverance (MOXIE) | 0.017 m³ | 300 W | ~2 years | 800 °C operational −60 °C ambient | $CO_2$ input $CH_4$ output |
| Gateway (HALO) | >125 m³ planned internal volume | ~60 kW | >2 years | -18 °C | Pressurized cabin air |
| Mars Hab (6 Crew) | 300 m³ | ~100 kW | 600 day nominal, 619 day maximum | -18 °C | Pressurized cabin air |

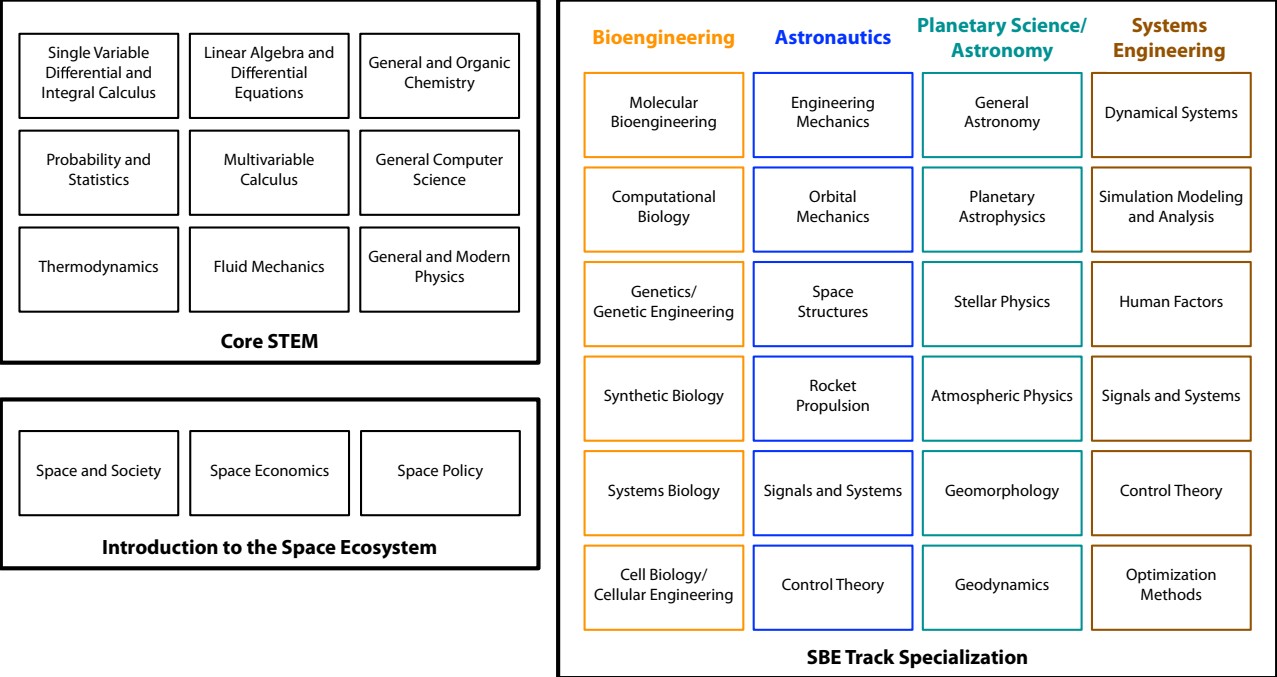

**Fig. 3 Conceptual undergraduate SBE program.** The SBE program is broken in three sections: core STEM (Science-Technology-Engineering-Mathematics) courses, introduction to space ecosystem courses, and track specialization courses for tracks in bioengineering, astronautics, planetary science & astronomy, and systems engineering.

limiting the end-product downstream analysis and use as feed-stocks for other integrated processes, as is needed to advance TRL beyond 6. This also cuts down on the ability to run DBTL diagnostics and SBE performance metrics on the system *in toto* as recyclability and sustainability are reliant on those end-products, and supportability if the processing is often reliant on Earth resources. Though much of the potential testing: polymerase chain reaction[46], imaging, and DNA sequencing[47,48] is possible with current miniaturized ISS modules, it may not all be at the scale needed for future experiments, and there may be gaps in capability as the field matures. Improved in situ data analysis through the development of new, high-throughput instruments could help close those gaps[49] and allow better metricization of whole systems under these new performance paradigms.

Lunar and Martian gravity is likely to have distinct biological effects compared to Earth gravity, resource composition, and radiation profile—and the ISS has only a limited volume in which to simulate them[50]. Additionally, both ambient environmental and target temperature windows span an extensive range across extraterrestrial environments, as do gas compositions, making representative testing more difficult in growth and testing chambers (plant, animal, and microbial) without full environmental control (Table 1). Environmental Control and Life Support System (ECLSS) systems for large-scale plant science requisite for advancing TRL for downstream lunar and Martian missions also require larger volume bounding boxes than is currently provided on the ISS[51]. Here we note the trade-offs with the tight volume and power stores on board. Smaller satellite modules can get technologies off the ground to advance TRL[52–54], but feature even greater size handicaps, and may prevent testing at the integrated, factory level in the DBTL cycle[55,56]. Scientific instruments and modules on rovers have been geared primarily for exploration and observation, not technology validation. Dedicated rovers or simply landing SBE payloads onto extra-terrestrial sites, SBE-ready orbiters, and Artemis operations as a stepping-stone to Mars can all demonstrate technology within a representative context and stand as some of the premier testbeds to flight qualify SBE prototypes[39]. In situ testing is key to the proposed SBE performance metrics: it forces technology and bioprocesses into accurate, integrated environments, and provides better confidence under radiation, microgravity, and isolation.

## Training of SBE minds

Maturation of space bioprocess engineering requires specialization of the training needed to produce the next generation of spacefaring scientists, engineers, astronauts, policy makers, and support staff[57]. Lessons learned from the Space Transportation System era led to calls for an increase in Science-Technology-Engineering-Mathematics (STEM) educational programs[58] beginning in secondary schools[59] and propagating to novel astronautics-based undergraduate[60] and graduate programs, and to the establishment of specialty space research centers focused on technology transfer[61]. The calls for workforce development were repeated just prior to the collapse of the Space Transportation System program, noting the dangers likely to arise from the lack of educational and training resources for those entering the space industry.[62]. Such a risk as described is especially poignant in the case of space-based biotechnologies given that mature technologies are far fewer, the new applications more futuristic, and the disciplines are not well represented in the traditional physics and engineering curricula. The Universities Space Research Association (USRA) lists 114 institutions with Space Technologies/Science academic programs while recent accounting of bioastronautics programs numbers 36[63]. However, the intersection between these lists yields only 22 schools. Given that US News names 250 world schools that have tagged themselves with Space Science programs, only ~8% of these are currently offering bioastronautics specialization—demonstrating that efforts that integrate human performance, life support, and bioengineering are under-served. Furthermore, the bioastronautics programs such as those offered by schools like

Harvard-MIT, University of Colorado Boulder, and Baylor University are not focused on biomanufacturing aspects that underlie SBE[64].

Academia must be prepared to capitalize on the opportunities of future SBE applications starting with either the creation of new and interdisciplinary programs or by assembling those from related disciplines (Fig. 1a). Because scientific and mathematical core courses are relatively standard across SBE-related disciplines, an effective foundation of technical skills could be easily constructed from the shared curriculum (Fig. 3). From there, specific SBE-driven training can be offered in (1) effects of space on plant and microbes; (2) process design for low gravity/high radiation; (3) management and storage of biological materials in space-based operations; (4) low energy/low mass bioreactor/bioprocessor design; (5) integrated biological systems engineering; (6) biological mission planning and logistics; (7) risk and uncertainty management; (8) containment and environmental impact of biological escape, films, corrosion, and cleanup; (9) policy awareness/development; and (10) ethics of cultivation and deployment. While the logistics for organizing such pathways for formal SBE training are non-trivial within the academic machine, we note that nearly all schools listed by USRA offer the component programs in bioengineering, planetary science or astronomy, and electrical or systems engineering. Since the courses for such engineering programs are standardized[65], it stands to reason that establishing focused SBE programs can begin by collecting and highlighting course combinations. As programs grow, additional faculty with SBE-driven research can be sourced. Such openings offer a much-needed opportunity to address systemic issues of diversity, equity, and inclusion both within SBE-based academia and the industrial space community at large[66].

## Moving forward

Making progress on the program above requires scientists, engineers, and policy experts to work together to verify, open, and update campaign specifications. The science requires scientists from multiple disciplines spanning biological and space systems engineering that require a degree of modularity, small footprints, and robustness not found elsewhere. Additionally, bioprocess and biological engineering must be applied to the building of cross-compatible and scalable processing systems and optimized organisms within the confines of space reactor and product. Finally, coordination mission specialists are critical to deploy tests into space during the run-up and through crewed missions. We argue that such groundwork requires multidisciplinary centers that can build long term partnerships and understanding; train the workforce in this unique application space; and perform the large-scale, long-term science necessary to succeed.

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

## Acknowledgements

This work is supported by the grant from the National Aeronautics and Space Administration (NASA, award number NNX17AJ31G).

## Author contributions

A.J.B., A.M., J.M.H., and A.P.A. conceived the concept based on the Center for the Utilization of Biological Engineering in Space (CUBES). D.H. led the graphics effort with assistance from A.J.B. G.M., I.L., N.J.H.A., A.A.M., A.M., and A.P.A. contributed to research and analyses. All authors (A.J.B., I.L., D.H., J.M.H., G.V., G.M., M.J.M., K.Y., N.J.H.A., C.S.C., T.W., L.C.S., C.S.C., S.M., K.A.M., A.A.M., A.M., and A.P.A.) wrote and edited the manuscript.

## Competing interests

The authors declare no competing interests.
