## [Peer Review File · Communications Engineering]

Reviewers' comments:

Reviewer #1 (Remarks to the Author):

The manuscript defined and discussed the scope and technology development cycle of Space Bioprocess Engineering (SBE). Furthermore, Design-Build-Test-Learn (DBTL) cycle for metabolic engineering was used for SBE Metrics and Methods description, which were of importance for develop the critical technologies for space missions. The following minor comments need to be addressed.

1. The bioproduct isolation and purification process usually need more procedures than fermentation in bioreactor, which needs to be specified and discussed more in the Space Bioprocess Engineering DBTL cycle.
2. The flexibility and various applicability are advantages for bioproduction and application for space mission, however, the biocontainment for engineered organisms needs to be considered for avoid unexpected issues (PMID: 25607366, PMID: 33033266, PMID: 33411536).
3. Cell-free bioproduction system is of importance for biotransformation and point-of-care biosensing, which needs to be discussed as part of the engineering system.

Reviewer #2 (Remarks to the Author):

Berliner et al present a perspective article on the need to prioritise the development of an emerging area 'Space Bioprocessing Engineering (SBE)' as a discipline in order to enable lunar and mars habitation.

I am in fierce agreement with the philosophies outlined and the approaches suggested. In particular, the rigorous comparison of SBE to conventional engineering solutions, new platforms to test and develop SBE solutions, training the next generation, and creating a global community of practice.

I have no significant concerns, only suggestions on improving clarity through some edits to syntax, to improve the flow of the article.

Minor corrections

Between L17-18 – 'train the next generation spacefaring workforce – could scan better.

L19 (of) long-term crewed ; delete point; in (the) demonstration

L19-23 – sorry, this is very long and difficult to read, it can likely be split up to improve readability. Delete etc?

L24 – intersectional is not the correct word here.

L32 – these (SBE) systems

L33 – plants fix C, this is a product alone, it does not have to be used as a feedstock for something else – just nutrition itself. This paragraph could perhaps be re-written, also no mention of fungal systems?

L40 – Suggest re-writing...

Ultimately, we propose: a mandate to include...; specialization (of) the metrics...; further development of the means by which...and train a new generation of minds...

L47 – systems,

L48 – ‘none of these’ – please define these

L51 – systems in (an) integrated

L53 – why Previous?, surely this is forward facing?

L 59-60, difference (in space) of gravity... remove etc.

L63 – biotechnologies,...approaches,

L69 – delete basis

L90 – we (propose to) adopt...

L108 – choices

L144 – test bed and bedding are definitely words; this is the first time of have seen test-bedded though.

L158 – reached ; and (in) limited flight..

L158 – chasses? Space ballet? Suggest ‘experiments’ would be better?

L177 – suture? recommend replacing with ‘close’?

L179 – gravity (is likely to have) distinct...

L183 – define ECLSS

L212 – Include in list policy awareness/development

Figure 1 – consider rearrangement of 1b to be more visually appealing.

The article was enjoyable to read, and stating this the mandate will be a useful base resource.

Best Regards,

M. Gilliam
University of Adelaide

Reviewer #3 (Remarks to the Author):

This paper proposed the concept of space bioprocess engineering to test, evaluate and optimize biotechnology-based space technology research for future human exploration space missions. On this basis, a research system of synthetic biology Design-Build-Test-Learn cycle is proposed, including performance evaluation indicators involving various biological function modules, and a combination of ground-based experiments and on-orbit experiments for space missions. At the same time, it is recommended to increase course teaching in the field of space biotechnology in research institutions such as universities, so as to provide a large number of researchers for the subsequent implementation of SBE.

Although not supported by actual research work, as a idea of view, I suggest that the article be accepted.

Reviewer #1 (Remarks to the Author):

The manuscript defined and discussed the scope and technology development cycle of Space Bioprocess Engineering (SBE). Furthermore, Design-Build-Test-Learn (DBTL) cycle for metabolic engineering was used for SBE Metrics and Methods description, which were of importance for develop the critical technologies for space missions. The following minor comments need to be addressed.

We thank Reviewer 1 for their comments which we believe we have addressed as efficiently as possible. Our goal with this perspective paper was to provide an introduction to the concepts of SBE, as opposed to a complete rendering of all biotechnology tradeoffs in the new field at large. Given this guiding principle, we hope that the minor comments provided have been sufficiently addressed, beginning on line 106.

2. The flexibility and various applicability are advantages for bioproduction and application for space mission, however, the biocontainment for engineered organisms needs to be considered for avoid unexpected issues (PMID: 25607366, PMID: 33033266, PMID: 33411536).

We thank Reviewer 1 for this suggestion. We have added the following line to our initial discussion of space-specific constraints on page 3: "... and (10) further consideration and development of biocontainment of engineered organisms to prevent (or at least mitigate) unexpected dispersal of unwanted living systems in pristine or tightly controlled environments [24-26]."

1. The bioproduct isolation and purification process usually need more procedures than fermentation in bioreactor, which needs to be specified and discussed more in the Space Bioprocess Engineering DBTL cycle.
3. Cell-free bioproduction system is of importance for biotransformation and point-of-care biosensing, which needs to be discussed as part of the engineering system.

Again, we thank Reviewer 1 for this suggestion. We have updated our text accordingly from lines 116-125. We address both the bioproduct isolation beyond fermentation and the cell-free production.

Reviewer #2 (Remarks to the Author):

Berliner et al present a perspective article on the need to prioritise the development of an emerging area 'Space Bioprocessing Engineering (SBE)' as a discipline in order to enable lunar and mars habitation.

I am in fierce agreement with the philosophies outlined and the approaches suggested. In particular, the rigorous comparison of SBE to conventional engineering solutions, new platforms to test and develop SBE solutions, training the next generation, and creating a global community of practice.

I have no significant concerns, only suggestions on improving clarity through some edits to syntax, to improve the flow of the article.

We appreciate the kind feedback from Reviewer 2.

Minor corrections

Between L17-18 – 'train the next generation spacefaring workforce – could scan better.

Updated now to "...and suggest a means to train the next generation of the spacefaring workforce on the SBE advantages and capabilities."

L19 (of) long-term crewed ; delete point; in (the) demonstration

Fixed.

L19-23 – sorry, this is very long and difficult to read, it can likely be split up to improve readability. Delete etc?

While we appreciate that the original sentence was very long and difficult to parse, we believe that this block of text is important in introducing some of the primary motivation for the following paper. We have updated the text as "While there has been progress in demonstration and evaluation of these benefits for specific examples in this field such as for food production, and waste recycling, there is only just emerging possible consensus on the scope of the application of biosynthetic and biotransformative technologies to space exploration. Additionally, there is almost no formal definition of the scope, performance needs and metrics, and technology development cycle for these systems." We believe this is more understandable, while also in keeping with the original intent.

L24 – intersectional is not the correct word here.

Fixed.

L32 – these (SBE) systems

Fixed.

L33 – plants fix C, this is a product alone, it does not have to be used as a feedstock for something else – just nutrition itself. This paragraph could perhaps be re-written, also no mention of fungal systems?

We thank Reviewer #2 for this comment. We have updated the statement to include "fungal systems." Our goal with this perspective paper was to provide an introduction to the concepts of SBE, as opposed to a

complete rendering of all biotechnology platforms tradeoffs in the new field at large. Given this guiding principle, we hope that the minor comments provided have been sufficiently addressed..

L40 – Suggest re-writing...

Ultimately, we propose: a mandate to include...; specialization (of) the metrics...; further development of the means by which...and train a new generation of minds...

While we appreciate that the original sentence was very long and difficult to parse, we believe that this block of text is important in introducing the thesis and structure of the following paper. We have rewritten this line as “Ultimately, efforts must be mounted to: (1) update the *mandate* to include SBE as a tool for enabling human exploration; (2) specialize the *metrics and methods* that guide SBE technology life-cycle and development; (3) further develop the means by which SBE technologies are designed for ground-based testing and matured on offworld platforms (Fig. 1d); and (4) train the *minds* entering the spacefaring workforce to better understand and leverage SBE advantages and capabilities.” We believe this is more understandable, while also in keeping with the original intent.

L47 – systems,

Fixed.

L48 – ‘none of these’ – please define these

Updated to “ However, no such survey has coherently called for...”

L51 – systems in (an) integrated

Fixed.

L53 – why Previous?, surely this is forward facing?

Very nice (and embarrassing) catch, thank you. Fixed.

L 59-60, difference (in space) of gravity... remove etc.

Fixed.

L63 – biotechnologies,...approaches,

Thank you for pointing this out. The line now reads “The development of open, publicly accessible data and tools would enable rigorous comparison among biotechnologies and abiotic (physical and chemical) approaches, and across mission-scenarios of higher-fidelity.”

L69 – delete basis

The term “basis” is not present in line 69. Please advise.

L90 – we (propose to) adopt...

Fixed.

L108 – choices

Fixed.

L144 – test bed and bedding are definitely words; this is the first time of have seen test-bedded though.
Updated. The line now reads “This allows for the in-flight testing of bioreactors previously evaluated on Earth...”

L158 – reached ; and (in) limited flight..

Fixed.

L158 – chasses? Space ballet? Suggest ‘experiments’ would be better?

Fixed. And I for one would certainly enjoy a low-gravity performance of the Ballet Firebird!

L177 – suture? recommend replacing with ‘close’?

Fixed.

L179 – gravity (is likely to have) distinct...

Fixed.

L183 – define ECLSS

Fixed. Environmental Control and Life Support System (ECLSS)

L212 – Include in list policy awareness/development

Added.

Figure 1 – consider rearrangement of 1b to be more visually appealing.

We appreciate this comment, and we have tried several alternatives. Each time we try a new version, we find that the figure becomes more “busy.” At this point we feel that Figure 1 is complete in its balance of aesthetics and information delivery, and we ask that this be left as is.

The article was enjoyable to read, and stating this the mandate will be a useful base resource.

We thank Reviewer 2 for his very helpful comments and suggestions!

Reviewer #3 (Remarks to the Author):

This paper proposed the concept of space bioprocess engineering to test, evaluate and optimize biotechnology-based space technology research for future human exploration space missions. On this basis, a research system of synthetic biology Design-Build-Test-Learn cycle is proposed, including performance evaluation indicators involving various biological function modules, and a combination of ground-based experiments and on-orbit experiments for space missions. At the same time, it is recommended to increase course teaching in the field of space biotechnology in research institutions such as universities, so as to provide a large number of researchers for the subsequent implementation of SBE.

Although not supported by actual research work, as a idea of view, I suggest that the article be accepted.

We appreciate the kind feedback from Reviewer 3, and we thank them for their comment!

REVIEWERS' COMMENTS:

Reviewer #1 (Remarks to the Author):

I suggest that the article be accepted.